

# Anxiety-like behavior and whole-body cortisol responses to components of energy drinks in zebrafish (*Danio rerio*)

Alia O. Alia and Maureen L. Petrunich-Rutherford

Department of Psychology, Indiana University Northwest, Gary, IN, USA

## ABSTRACT

The current study investigated the independent and combined effects of caffeine and taurine on anxiety-like behavior and neuroendocrine responses in adult zebrafish (*Danio rerio*). Caffeine (1,3,7-trimethylpurine-2,6-dione), the world's most commonly used psychoactive drug, acts as an adenosine receptor blocker and a mild central nervous system stimulant. However, excessive use of caffeine is associated with heightened anxiety levels. Taurine (2-aminoethanesulfonic acid), a semi-essential amino acid synthesized within the human brain, has been hypothesized to play a role in regulating anxiolytic behavior. Caffeine and taurine are two common additives in energy drinks and are often found in high concentrations in these beverages. However, few studies have investigated the interaction of these two chemicals with regards to anxiety measures. A suitable vertebrate to examine anxiety-like behavior and physiological stress responses is the zebrafish, which has shown promise due to substantial physiological and genetic homology with humans. Anxiety-like behavior in zebrafish can be determined by analyzing habituation to novelty when fish are placed into a novel tank and scototaxis (light avoidance) behavior in the light-dark test. Stress-related neuroendocrine responses can be measured in zebrafish by analyzing whole-body cortisol levels. The goal of this study was to determine if exposure to caffeine, taurine, or a combination of the two compounds altered anxiety-like behavior and whole-body cortisol levels in zebrafish relative to control. Zebrafish were individually exposed to either caffeine (100 mg/L), taurine (400 mg/L), or both for 15 min. Zebrafish in the control group were handled in the same manner but were only exposed to system tank water. After treatment, fish were transferred to the novel tank test or the light-dark test. Behavior was tracked for the first 6 min in the novel tank and 15 min in the light-tark test. Fifteen min after introduction to the behavioral task, fish were euthanized for the analysis of whole-body cortisol levels. The results demonstrate that caffeine treatment decreased the amount of exploration in the top of the novel tank and increased scototaxis behavior in the light-dark test, which supports the established anxiogenic effect of acute exposure to caffeine. Taurine alone did not alter basal levels of anxiety-like behavioral responses nor ameliorated the anxiogenic effects of caffeine on behavior when the two compounds were administered concurrently. None of the drug treatments altered basal levels of whole-body cortisol. The current results of this study suggest that, at least at this dose and time of exposure, taurine does not mitigate the

Corresponding author
Maureen L. Petrunich-Rutherford,
mlpetrun@iun.edu

anxiety-producing effects of caffeine when administered in combination, such as with energy drink consumption.

# INTRODUCTION

Caffeine (1,3,7-trimethylpurine-2,6-dione), the world's most widely consumed psychoactive drug, has stimulant-like effects on the central nervous system and overall behavior (*Evans & Battisti, 2018*). Widespread use of caffeine is likely due to the positive effects it has on increasing mental alertness and physical endurance as well as reducing fatigue and overall tiredness (*Heckman, Weil & De Mejia, 2010*). However, caffeinated beverages may also be associated with increasing anxiety and other negative health outcomes (*Richards & Smith, 2016*), particularly in youth (*De Sanctis et al., 2017*) and in individuals with certain genetic variants associated with caffeine pharmacokinetics and pharmacodynamics (*Nehlig, 2018*). The consumption of energy drinks has increased significantly over the last decade in all age groups surveyed in a recent study (*Vercammen, Koma & Bleich, 2019*). It is likely that energy drinks are popular due to the stimulant-like effects produced by caffeine; thus, energy drinks are commonly consumed by populations such as young adults in college settings for supporting academic demands (*Trunzo et al., 2014*) and by military service members (*Attipoe et al., 2018*). However, as with beverages containing caffeine, excessive consumption of energy drinks is associated with negative health outcomes like adverse cardiac events (for review, see *Higgins et al., 2018*). Adolescents are more at risk to experiencing ill effects of energy drink consumption (*Curran & Marczinski, 2017*). In addition, energy drinks are frequently consumed with alcohol, a practice which is associated with increased risky decision making (*Manchester, Eshel & Marion, 2017*) and increased risk for negative health consequences in young adults (*Caviness, Anderson & Stein, 2017*).

Although caffeine is one of the primary chemicals present in energy drinks, another additive found in high concentrations is taurine. Taurine (2-aminoethanesulfonic acid) is considered a semi-essential amino acid, as is it not used in protein synthesis (*Ripps & Shen, 2012*). However, taurine does play other critical roles, particularly in the central nervous system, such as by helping to regulate cell volume and modulate neurotransmission in the brain (*Oja & Saransaari, 2017*). In addition, taurine has been hypothesized to play a role in anxiolytic behavior in rodents (*Chen et al., 2004*; *El Idrissi et al., 2009*; *da Silva Francisco & Guedes, 2015*; *Kong et al., 2006*; *McCool & Chappell, 2007*; *Wu et al., 2017*; *Zhang & Kim, 2007*). The effects of caffeine and taurine in combination have mainly been studied in the context of physical performance and cardiovascular function. Administration of caffeine and taurine in combination altered measures of cardiovascular function (*Bichler, Swenson & Harris, 2006*) and elevated mental performance and mood (*Seidl et al., 2000*) over placebo in human participants. When these two components were studied alone

and in combination, taurine counteracted the effects of caffeine on cardiovascular function (*Schaffer et al., 2014*), mitigated some of the effects of caffeine on cognitive measures (*Giles et al., 2012*), reduced caffeine-induced physiological alterations associated with cycling performance (*Warnock et al., 2017*), and attenuated the effects of caffeine on specific parameters of reaction time (*Peacock, Martin & Carr, 2013*) in human subjects. However, *in vitro*, taurine did not alter caffeine-induced effects in human cardiac muscle tissue (*Chaban et al., 2017*) or mouse skeletal muscle tissue (*Tallis et al., 2014*). In other measures in a variety of animal models, caffeine and taurine have synergistic effects, such as on sleep parameters in *Drosophila* (*Lin et al., 2010*), plasma calcium levels in rats (*Owoyele et al., 2015*), locomotor activity in mice (*Kimura et al., 2009*), and memory and attention in rats (*Valle et al., 2018*). Thus, the specific impact of caffeine and taurine appears to be dependent on the physiological and behavioral parameters under investigation. Although caffeine and taurine can modulate anxiety-like states on their own, little is known regarding the impact of these two popular energy drink components in combination.

The zebrafish (*Danio rerio*) animal model is rapidly becoming an attractive model organism in neuropharmacology research due to its low cost and ease of maintenance; in addition, the nervous and endocrine systems regulating biological and behavioral responses to stress are highly conserved (*Stewart et al., 2012*). Stress and anxiety-like states can be inferred in the zebrafish model by measuring various behavioral responses such as shoaling, immobility, erratic movements, and the detection of jumping in response to various stimuli such as the introduction of pharmaceuticals, visual stimuli, and alarm pheromones (*Egan et al., 2009*; *Maximino et al., 2014*; *Wong et al., 2010*). The novel tank test is a well-validated measure of anxiety-like behavior in zebrafish and involves measuring freezing and exploratory behavior upon introduction to a new tank (*Kysil et al., 2017*; *Mezzomo et al., 2016*; *Raymond et al., 2012*). Another behavioral paradigm, the light-dark test, measures anxiety-like behavior in the form of scototaxis, or light avoidance, in response to pharmacological or behavioral manipulation (*Maximino et al., 2010*; *Stewart et al., 2011*). Stress responses can also be assessed through measuring neuroendocrine responses, namely cortisol, elicited by specific stimuli (*Cachat et al., 2010*; *Canavello et al., 2011*). The hypothalamic-pituitary-interrenal (HPI) axis of teleost species, such as zebrafish, is homologous to the hypothalamic-pituitary adrenal (HPA) axis of mammals (*Nesan & Vijayan, 2013*; *Wendelaar Bonga, 1997*). Thus, exposure to pharmacological compounds will elicit behavioral and neuroendocrine effects in the fish that may generally model the effects these compounds have in humans.

Consistent with findings in rodent models and human subjects, zebrafish exposed to caffeine at a variety of ages demonstrate anxiety-like behavior (*Egan et al., 2009*; *Richendrfer et al., 2012*; *Rosa et al., 2018*; *Schnörr et al., 2012*; *Steenbergen, Richardson & Champagne, 2011*; *Wong et al., 2010*). The specific anxiogenic effect of caffeine is possibly due to antagonism at $A_1$ adenosine receptors (*Maximino et al., 2011*). Acute exposure to taurine, on the other hand, is associated with anxiolytic effects on behavior (*Mezzomo et al., 2016*). The modulation of anxiety-like behavior by taurine in zebrafish may be induced by blunting neuroendocrine cortisol responses to stress (as observed in *Mezzomo et al., 2019*). Similarly, glucocorticoid hormone regulation has been proposed as a possible

mechanism by which taurine modulates anxiety-like behavior in unpredictable chronic stress exposure in rodents (*Wu et al., 2017*). However, whether taurine mitigates the anxiogenic effects of caffeine via HPA/HPI regulation is currently not known.

Thus, in this study, the effects of caffeine and taurine on anxiety-like behavior and neuroendocrine responses were explored. The purpose of this study was to determine if acute exposure to caffeine, taurine, or both altered anxiety-like behavior and whole-body cortisol levels in zebrafish. If caffeine operates as an anxiogenic in zebrafish as expected, then the fish will exhibit more anxiety-like behavior and display increased whole-body cortisol levels relative to control. If taurine operates as an anxiolytic in zebrafish, then the fish exposed to taurine should have decreased cortisol levels and exhibit decreased anxiety-like behavior, such as spending more time in the upper portion of the tank during the novel tank test or entering the light zone more frequently in the light-dark test. If taurine modulates the effects of caffeine, taurine should mitigate any caffeine-induced anxiety-like effects on behavior and increases in whole-body cortisol when the fish are acutely exposed to both drugs simultaneously. This study will potentially aid in elucidating the effects of caffeine and taurine when co-administered acutely. In addition, the findings from this study will provide insight on the interaction between chemicals commonly found in energy drinks and whether the modulation of anxiety-like behavior is related to the activity of the neuroendocrine stress axis.

## METHODS AND MATERIALS

### Animals and housing

Wild-type, adult zebrafish ($N = 139$) were purchased from Carolina Biological Supply (Burlington, NC, USA). Upon arrival to the facility, the fish were maintained on a circulating system on a 14:10 light:dark cycle at a density of approximately five to six fish per liter. Fish were fed flake food once per day and dried brine shrimp once per day. The internal environment of the housing tanks was maintained at a temperature of $26 \pm 2$ °C. Animals were housed and maintained in accordance with ethical guidelines (*Harper & Lawrence, 2016*; *National Research Council, 2011*; *Westerfield, 2000*). All fish were allowed to acclimate to the facility for at least a week before any experimental procedures were conducted (*Dhanasiri, Fernandes & Kiron, 2013*). All experimental procedures involving animals were performed between 9:00 a.m. and 1:00 p.m.

### Drug administration

Drugs were purchased from Santa Cruz Biotechnology (Dallas, TX, USA). On the day of the experiment, housing tanks were removed from the system and placed in the experimentation room 30 min prior to treatment to allow for habituation. Individual fish were selected at random, carefully netted from the tank, and placed in a tank containing one L of either a drug or control solution for a duration of 15 min. There were four independent conditions: control (system water), caffeine (1,3,7-trimethylpurine-2,6-dione), taurine (2-aminoethanesulfonic acid), or a combination of caffeine/taurine. Fish were either immersed in a solution of either caffeine ($N = 29$) at 100 mg/L (*Maximino et al., 2014*), taurine ($N = 30$) at 400 mg/L (*Mezzomo et al., 2016*), or caffeine and taurine

combined ($N$ = 29) at 100 and 400 mg/L, respectively. Subjects in the control group ($N$ = 51) were simply immersed in system water for 15 min. Treatment solutions were replaced after every subject. Immediately following treatment, each subject was transferred to either a novel tank (Experiment 1) or light-dark tank (Experiment 2) for behavioral analysis.

## Novel tank test (experiment 1)

After drug treatment, individual fish ($N$ = 100 total) were placed in a trapezoidal tank (15.2 cm height × 27.9 cm top × 22.5 cm bottom × 7.1 cm width). The tank was positioned to allow for recording of behavior from the wide side of the tank, using a camera placed on a tripod. The first 6 min of behavior was recorded and subsequently analyzed by Ethovision XT software (Noldus, Leesburg, VA, USA), which was generously provided as part of the Faculty for Undergraduate Neuroscience Equipment Loan Program. Behavioral measures included the total distance traveled (cm), mean speed during ambulation (cm/s), immobility duration (s), number of times fish were immobile, latency to first top entry (s), total time in top (s), distance in top (cm), and number of entries to top. The percentage of fish from each group that did not re-enter the top zone after being introduced to the novel tank ($N$ = 18 total) was also calculated. These samples were not included in the analysis of the latency to top measure but were included in all other behavioral measures. One subject was not included in behavioral analyses due to corruption of the video file.

## Light-dark test (experiment 2)

After drug treatment, individual fish ($N$ = 39 total) were placed in a rectangular tank (approximately 15 × 30 × 20 cm) with a water depth of three cm. The dark side of the tank (sides and bottom) was covered with black plastic aquarium background and the other side was left uncovered (*Magno et al., 2015*; *Maximino et al., 2010*). The behavior of the fish was recorded from above the tank for 15 min with a Logitech C922x Pro Stream Webcam. Video files were uploaded to and analyzed with BehaviorCloud motion-tracking software (https://www.behaviorcloud.com/, San Diego, CA, USA). Six behavioral measures were quantified for each fish: total distance traveled (cm), mean speed during ambulation (cm/s), immobility duration (s), number of entries to the light zone, total time spent in light zone (min), and total distance traveled in the light zone (cm).

## Euthanasia

Fifteen min after each subject was introduced to the respective behavioral task, the fish were netted from the tank and placed into 30 mL euthanasia solution (0.1% clove oil in system water) for approximately 60 s (*Davis et al., 2015*; *Wong et al., 2014*). Once the subjects displayed no movement or responsiveness, the bodies were gently dried, placed in a microcentrifuge tube, and were stored at −20 °C until the cortisol extraction was performed.

## Cortisol extraction/assay

Cortisol extraction and analysis procedures were adapted from previously published methods (*Cachat et al., 2010*; *Canavello et al., 2011*). Briefly, whole-body samples were

thawed and individually weighed. Samples were cut into smaller pieces with a scalpel and placed in one mL of 25 mM of ice-cold phosphate buffered saline (PBS). Each sample was homogenized for 30–60 s and placed back on ice. Diethyl ether (five mL) was added to each sample and thoroughly vortexed, then centrifuged at 2,500×$g$ rpm for 15 min. Following the centrifugation, the organic layer containing the cortisol was removed from the sample and placed in a separate tube. The addition of ether, vortexing, centrifugation, and organic layer removal was repeated two more times to maximize the amount of cortisol extracted from each sample. The samples from Experiment 1 were allowed to dry at room temperature under a fume hood until the ether layer was fully evaporated; samples from Experiment 2 were dried with a light stream of air. In both procedures, samples were dried until a yellow oil containing cortisol remained.

After the samples were dry, one mL of ice-cold PBS was added to each tube, and a commercially-available enzyme-linked immunosorbent assay (ELISA) kit (Salimetrics, Carlsbad, CA, USA) was used to assess cortisol levels. ELISA procedures were conducted according to manufacturer instructions. Binding values for each sample was compared to a standard curve generated by My Curve Fit software (https://mycurvefit.com/). Cortisol levels were normalized to body weights of each sample and are displayed in ng cortisol/g body weight. Four samples from Experiment 1 were excluded from the cortisol analysis due to methodological errors incurred during the extraction procedure.

## Data analysis

Behavioral and cortisol dependent measures were expressed as the mean ± standard error of the mean and were analyzed using a two-way analysis of variance (ANOVA) with caffeine (levels: yes, no) and taurine (levels: yes, no) as the independent variables. Group means were compared by a Tukey post hoc test when appropriate. The percentage of each group that did not explore the top zone of the novel tank test was analyzed with a Chi-squared test. All analyses were conducted using JASP software (https://jasp-stats.org/). Results were considered statistically significant if $p < 0.05$.

## RESULTS

Anxiety-like responses of subjects were determined by assessing behavioral measures exhibited within the novel tank test and light-dark test. In addition, whether neurochemical measures of anxiety were altered by drug treatment was assessed by measuring whole-body cortisol levels after each of the behavioral tasks. Measures of each behavioral test were broken down into three discrete domains: motor activity, immobility, and exploration. In the novel tank test (Experiment 1), motor activity was assessed by examining the total distance moved overall and the mean speed of the subjects (while mobile) within the tank. The second domain assessed the number of times the subjects were immobile and the total duration of immobility (s). The third domain included the activity in the top zone of the novel tank; this included assessing the distance moved in the top zone (cm), the number of times the subjects entered the top zone, the time spent in the top zone (s), and the latency to first top entry (s). The percentage of subjects in each group that did not explore the top zone was also calculated. In the light-dark test

(Experiment 2), motor activity was represented by the total distance moved overall and the mean speed of the subjects (while mobile) within the tank. The second domain assessed freezing by measuring the duration of immobility (s). The third domain included the activity in the light zone of the tank; this included assessing the number of times the subjects entered the light zone, the distance moved in the light zone (cm), the time spent in the light zone (s), the number of crossings from one compartment to the other, and the latency to re-enter the light zone after the first visit to the dark zone (s).

## Experiment 1

### Motor activity in the novel tank test

The total distance traveled and the mean ambulatory speed in the novel tank test is illustrated in Fig. 1. A two-factor ANOVA revealed no significant main effect of caffeine ($F(1,95) = 2.898$, $p = 0.092$), no significant main effect of taurine ($F(1,95) = 1.101$, $p = 0.297$), and no significant interaction between caffeine and taurine ($F(1,95) = 2.263$, $p = 0.136$) on the total distance traveled in the novel tank test (Fig. 1A). A two-factor ANOVA indicated a marginally significant main effect of caffeine ($F(1,95) = 3.214$, $p = 0.076$), no significant main effect of taurine ($F(1,95) = 0.061$, $p = 0.806$), but no significant interaction between caffeine and taurine ($F(1,95) = 0.047$, $p = 0.829$) on the mean ambulatory speed traveled in the novel tank test (Fig. 1B). Thus, it appears that caffeine and taurine, either alone or in combination, did not significantly affect general motor activity of adult zebrafish in the novel tank test.

### Freezing behavior in the novel tank test

Freezing behavior (Fig. 2) displayed in zebrafish in the novel tank test can be used as an indication of anxiety-like behavior induced by treatment. As the number of freezing bouts or time spent immobile increases, it can be inferred that the subjects are experiencing higher levels of anxiety. A two-factor ANOVA revealed no significant main effect of caffeine ($F(1,95) = 1.674$, $p = 0.199$), no significant main effect of taurine ($F(1,95) = 0.534$, $p = 0.467$), and no significant interaction between caffeine and taurine ($F(1,95) = 0.339$, $p = 0.562$) on the number of immobility bouts in the novel tank test (Fig. 2A). A two-factor ANOVA indicated no significant main effect of caffeine ($F(1,95) = 1.004$, $p = 0.319$), no significant main effect of taurine ($F(1,95) = 0.062$, $p = 0.805$), and no significant interaction between caffeine and taurine ($F(1,95) = 0.184$, $p = 0.669$) on the total time spent immobile in the novel tank test (Fig. 2B). Thus, it appears that caffeine and taurine, either alone or in combination, did not significantly affect freezing behavior of adult zebrafish in the novel tank test.

### Exploratory behavior in the novel tank test

Figure 3 displays the mean ± SEM for each group for each of the four exploratory measures of interest in the novel tank test. If the subjects are less exploratory (e.g., spend less time in the top, enter the top fewer times, etc.), then it can be inferred that the subjects are experiencing more anxiety. A two-factor ANOVA revealed no significant main effect of caffeine ($F(1,95) = 2.019$, $p = 0.159$), no significant main effect of taurine ($F(1,95) = 2.150$, $p = 0.146$), and no significant interaction between caffeine and taurine ($F(1,95) = 2.701$,

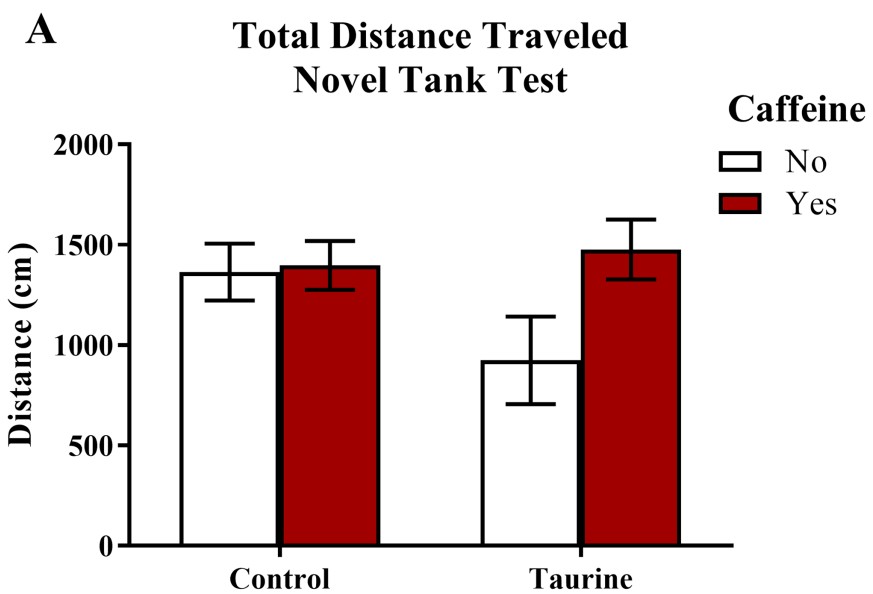

**A**

## Total Distance Traveled Novel Tank Test

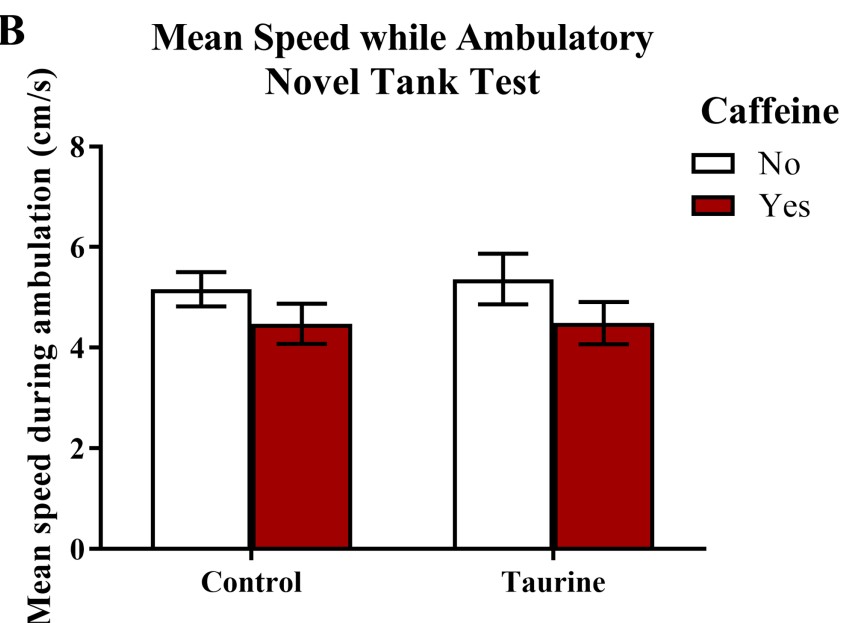

**B**

## Mean Speed while Ambulatory Novel Tank Test

**Figure 1 Measures of zebrafish motor activity in the novel tank test.** Acute exposure to energy drink components (caffeine, taurine, or both) did not alter (A) the total distance traveled and (B) the mean speed while ambulatory in the novel tank test in adult zebrafish. Bars indicate means of each group ±SEM.

$p = 0.104$) on the distance traveled in the top zone of the novel tank test (Fig. 3A). A two-factor ANOVA indicated a significant main effect of caffeine ($F(1,95) = 6.379$, $p = 0.013$, caffeine < no caffeine), no significant main effect of taurine ($F(1,95) = 0.515$, $p = 0.475$), but no significant interaction between caffeine and taurine ($F(1,95) = 0.021$, $p = 0.886$) on the number of entries to the top zone of the novel tank test (Fig. 3B).

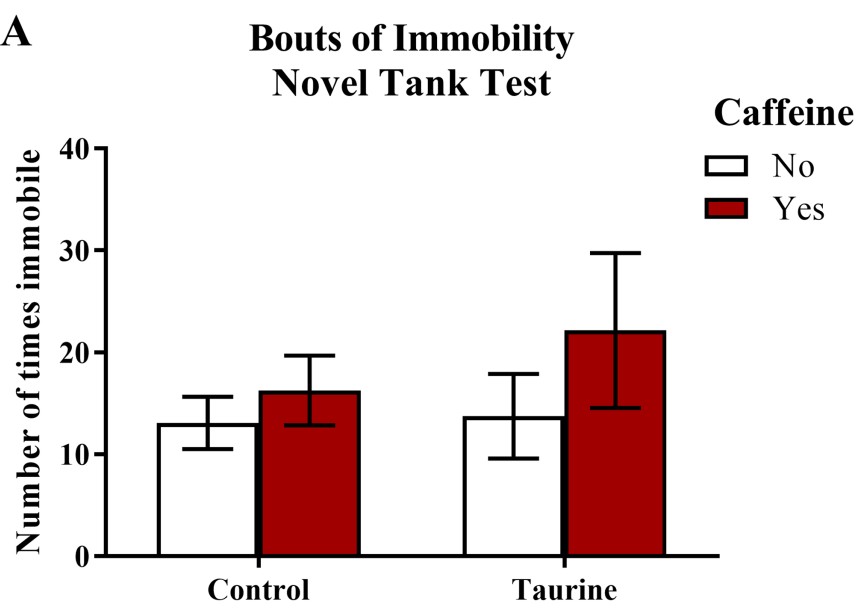

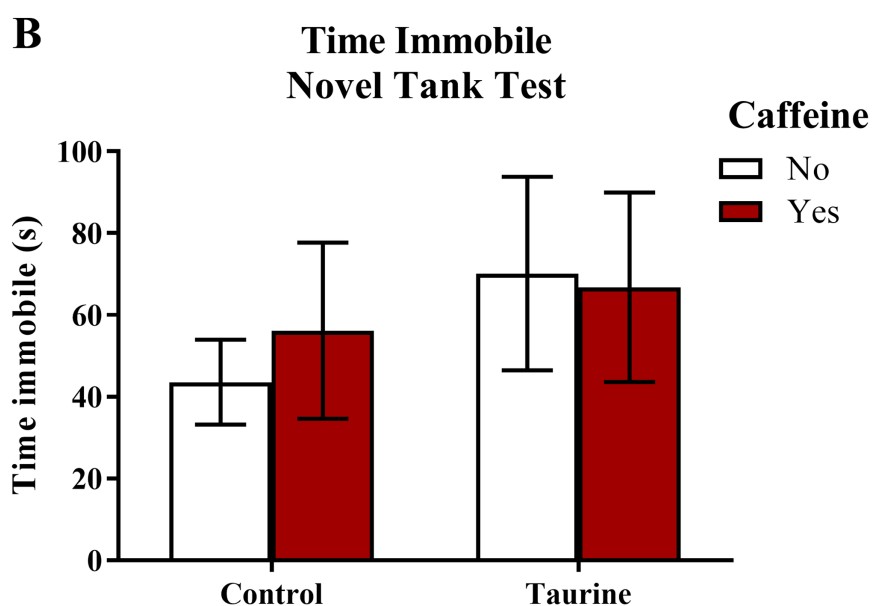

**Figure 2 Measures of zebrafish freezing behavior in the novel tank test.** Acute exposure to energy drink components (caffeine, taurine, or both) did not alter (A) the total number of immobility bouts and (B) the total time spent immobile in the novel tank test in adult zebrafish. Bars indicate means of each group ±SEM.

A two-factor ANOVA revealed no significant main effect of caffeine ($F(1,95) = 0.361$, $p = 0.550$), no significant main effect of taurine ($F(1,95) = 1.480$, $p = 0.227$), and no significant interaction between caffeine and taurine ($F(1,95) = 2.372$, $p = 0.127$) on the time spent in the top zone of the novel tank test (Fig. 3C). A two-factor ANOVA indicated no significant main effect of caffeine ($F(1,77) = 0.786$, $p = 0.378$), a significant main effect of taurine ($F(1,77) = 4.308$, $p = 0.041$, taurine < no taurine), but no significant

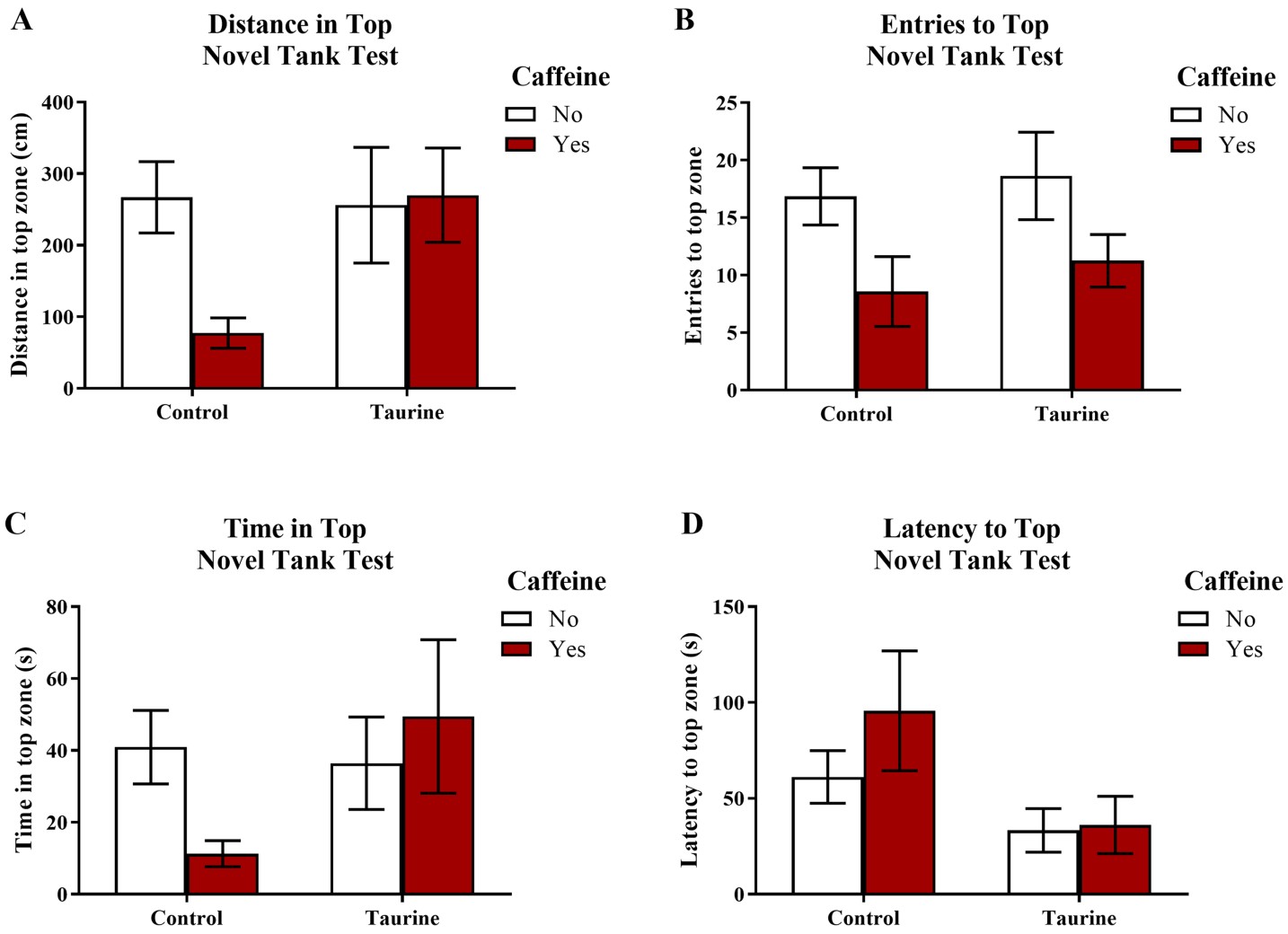

**Figure 3 Measures of zebrafish exploratory behavior in the novel tank test.** Acute exposure to energy drink components (caffeine, taurine, or both) altered exploratory behavior in the novel tank test in adult zebrafish. Caffeine decreased (A) the distance traveled in the top zone, (B) the number of entries to the top zone, and (C) the time spent in the top zone. Taurine decreased (D) the latency to enter the top zone. Bars indicate means of each group ±SEM.                     

**Table 1 Percentage of zebrafish that did not explore top zone in the novel tank test.**

| Variable | Control (N = 41) | Caffeine (N = 19) | Taurine (N = 19) | CAF+TAU (N = 20) | $\chi^2$ | $p$ |
|---|---|---|---|---|---|---|
| Did not explore top | 9.8% | 42.1% | 5.3% | 25.0% | 12.02 | 0.007 |

Note:
Acute exposure to energy drink components (caffeine, taurine, or both (CAF+TAU)) did significantly influence the number of fish that failed to explore the top zone in the novel tank test in adult zebrafish.

interaction between caffeine and taurine ($F(1,77) = 0.567$, $p = 0.454$) on the latency to enter the top zone of the novel tank test (Fig. 3D). It is of note that, for this behavioral task, not all fish returned to the top (see Table 1) and thus could not be included in this analysis. Table 1 displays the percentage of fish from each group that did not explore the top portion of the novel tank. Almost half (42.1%) of the caffeine-treated group did

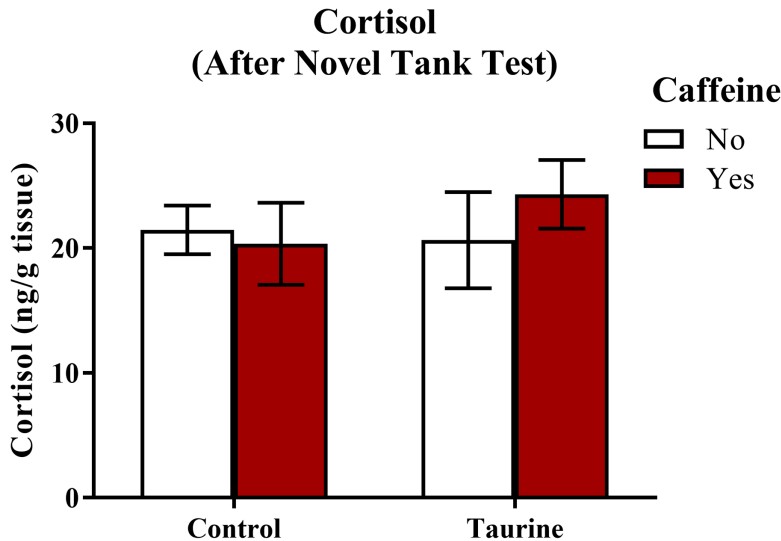

**Figure 4 Measures of zebrafish neuroendocrine function after the novel tank test.** Acute exposure to energy drink components (caffeine, taurine, or both) did not alter whole-body cortisol levels of zebrafish of fish in Experiment 1 (fish were sacrificed after the novel tank test). Bars indicate means of each group ±SEM.                                                                      

not explore the top but fewer than 10% of the fish exposed to the control or taurine conditions did not explore the top. About 25% of fish exposed to the mixed drug treatment failed to explore the top zone in the novel tank test. These group differences were significant according to a chi-squared test ($\chi^2$ (3, $N$ = 99) = 12.02, $p$ = 0.007). Thus, the pattern of data suggests that caffeine treatment decreased the tendency to explore the top, and that fish that did re-enter the top took more time to do so after treatment with caffeine alone.

Although some of the exploratory measures did not reach the criterion for statistical significance, in general, caffeine-treated fish appeared to demonstrate less exploratory behavior in the top zone, whereas taurine generally did not alter overall exploration besides shortening the latency to explore the top zone. The data loosely suggest that when caffeine and taurine were co-administered, taurine may have mitigated some of the effects of caffeine on exploration (e.g., increased distance and time spent in the top zone and decreased the latency to enter the top zone of the novel tank test); however, a higher dose and/or longer course of exposure to taurine is likely necessary to elicit any significant effect on caffeine-induced anxiety-like behavior in the novel tank test.

### Whole-body cortisol levels post-novel tank test

A neurochemical marker of anxiety was determined by analyzing whole-body cortisol levels of each subject (Fig. 4). A two-factor ANOVA revealed no significant main effect of caffeine ($F(1,92)$ = 0.189, $p$ = 0.665), no significant main effect of taurine ($F(1,92)$ = 0.283, $p$ = 0.596), and no significant interaction between caffeine and taurine ($F(1,92)$ = 0.660, $p$ = 0.419) on whole body cortisol levels (Fig. 4). Thus, it does not appear that acute exposure to different components of energy drinks altered stress hormone responses, at least when cortisol was assessed 15 min after introduction to the novel tank test.

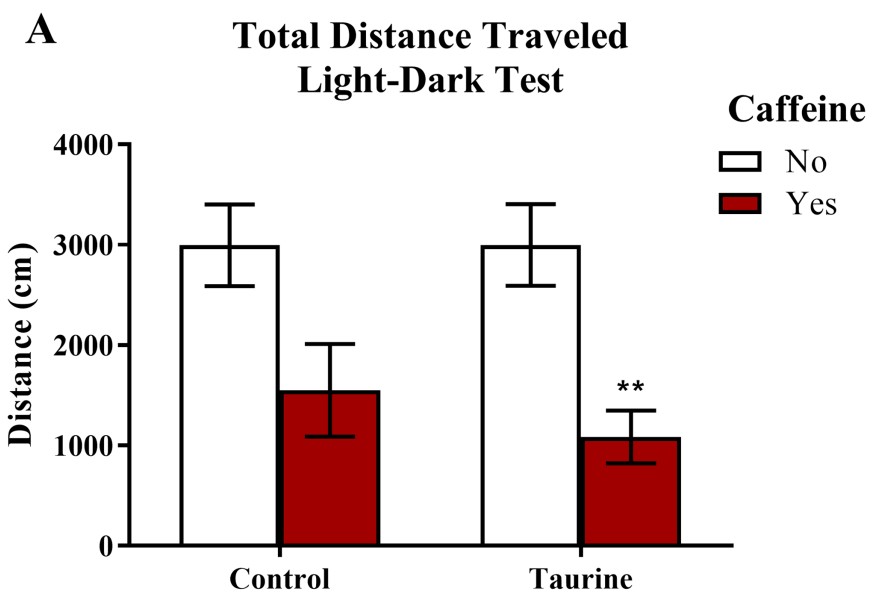

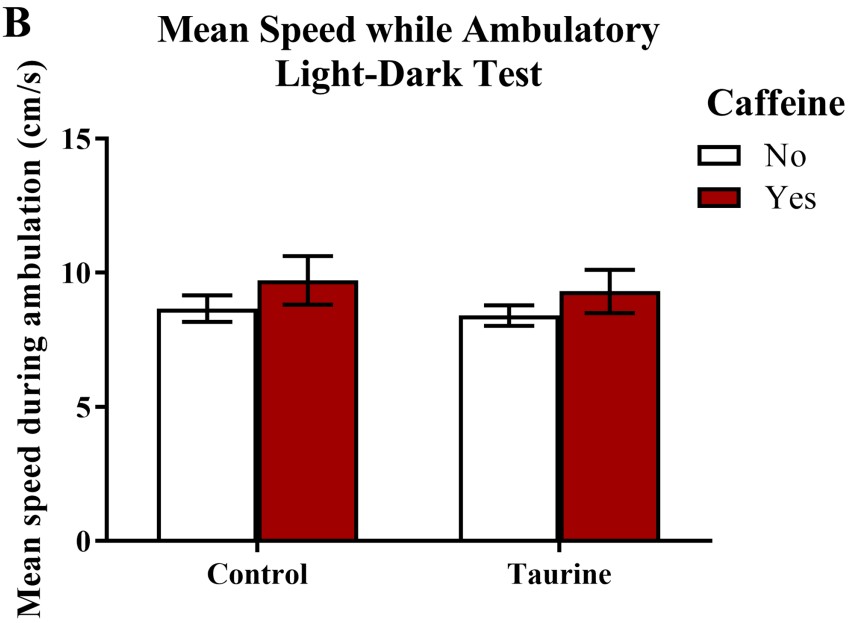

**Figure 5 Measures of zebrafish motor activity in the light-dark test.** Acute exposure to energy drink components (caffeine, taurine, or both) affected some aspects of motor activity in adult zebrafish. Acute caffeine decreased (A) the total distance traveled but not (B) the mean speed during ambulation in the light-dark test. Bars indicate means of each group ±SEM and ** indicates significant ($p < 0.01$) difference from respective non-caffeine treated control (Tukey post hoc).   

### Experiment 2

#### Motor activity in the light-dark test

Similar to the novel tank test, the total distance traveled and the mean ambulatory speed (Fig. 5) can be used as markers for general motor activity in the light-dark test.

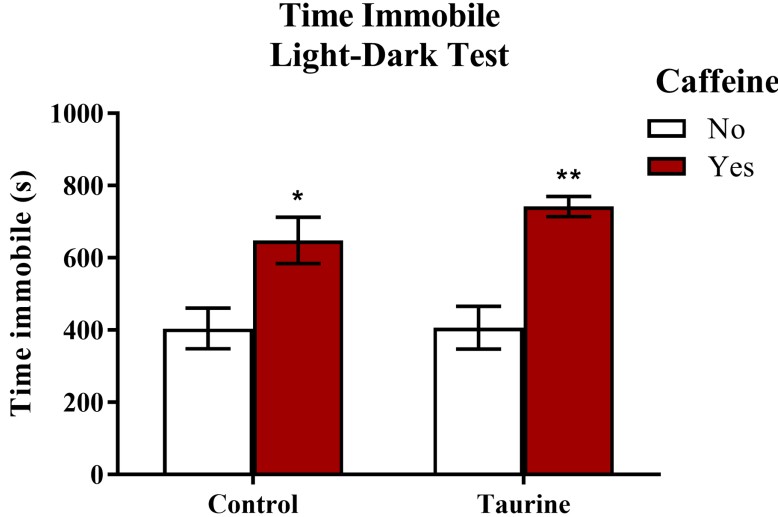

**Figure 6 Measures of zebrafish freezing behavior in the light-dark test.** Acute exposure to energy drink components (caffeine, taurine, or both) altered the total time spent immobile in the novel tank test in adult zebrafish. Caffeine increased the total time immobile in the light-dark test. Bars indicate means of each group ±SEM. *, ** indicates significant ($p < 0.05$ and $p < 0.01$, respectively) difference from respective non-caffeine treated control (Tukey post hoc).

A two-factor ANOVA revealed a significant main effect of caffeine ($F(1,35) = 17.791$, $p < 0.001$, caffeine < no caffeine), no significant main effect of taurine ($F(1,35) = 0.336$, $p = 0.566$), but no significant interaction between caffeine and taurine ($F(1,35) = 0.343$, $p = 0.562$) on the total distance traveled in the light-dark test (Fig. 5A). A two-factor ANOVA indicated no significant main effect of caffeine ($F(1,35) = 2.074$, $p = 0.159$), no significant main effect of taurine ($F(1,35) = 0.245$, $p = 0.624$), and no significant interaction between caffeine and taurine ($F(1,35) = 0.013$, $p = 0.911$) on the mean speed traveled (while ambulatory) in the light-dark test (Fig. 5B). Thus, it appears that caffeine, but not taurine, significantly decreased the total distance traveled by adult zebrafish in the light-dark test. None of the treatments altered mean swimming speed of the subjects.

### Freezing behavior in the light-dark test

Freezing behavior (Fig. 6) displayed in zebrafish can be used as an indication of anxiety-like behavior induced by treatment. A two-factor ANOVA indicated a significant main effect of caffeine ($F(1,35) = 27.792$, $p < 0.001$, caffeine > no caffeine), no significant main effect of taurine ($F(1,35) = 0.764$, $p = 0.388$), but no significant interaction between caffeine and taurine ($F(1,35) = 0.699$, $p = 0.409$) on the total time spent immobile in the novel tank test (Fig. 6). Thus, it appears that caffeine, but not taurine, significantly increased the time spent immobile by adult zebrafish in the light-dark test. This may explain why the total distance traveled was less for caffeine-treated fish (see Fig. 5A).

### Exploratory behavior in the light-dark test

Figure 7 displays the mean ± SEM for each group for each of the three exploratory measures of interest in the light-dark test. If the subjects are less exploratory (e.g., spend less time in the light zone, enter the light zone fewer times, etc.) in the light-dark test, then

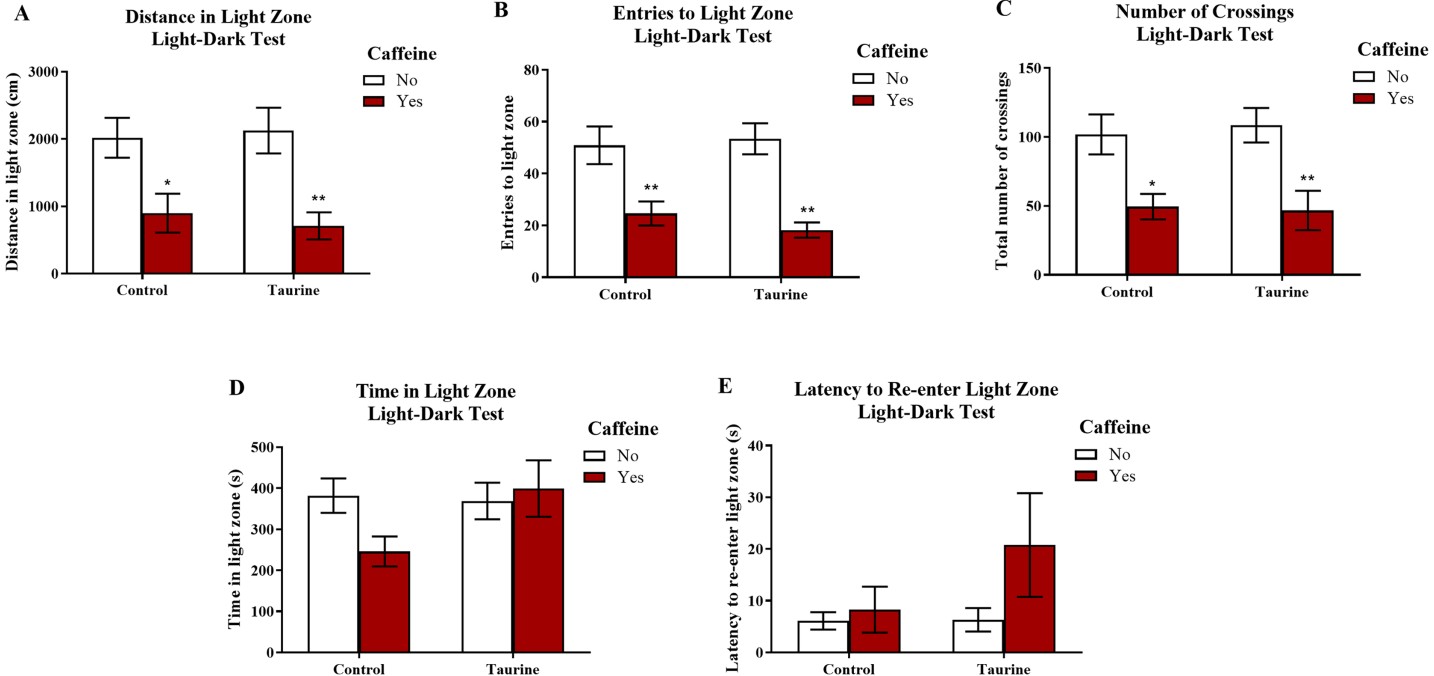

**Figure 7 Measures of zebrafish exploratory behavior in the light-dark test.** Acute exposure to energy drink components (caffeine, taurine, or both) altered exploratory behavior in the novel tank test in adult zebrafish. Caffeine decreased (A) the distance traveled in the light zone, (B) the number of entries to the light zone, and (C) the number of crossings between compartments. The acute drug treatments did not significantly alter (D) the time spent in the light zone nor (E) the latency to re-enter the light zone after the first visit to the dark zone. Bars indicate means of each group ±SEM. *, ** indicates significant ($p < 0.05$ and $p < 0.01$, respectively) difference from respective non-caffeine treated control (Tukey post hoc).

it can be inferred that the subjects are experiencing more anxiety. A two-factor ANOVA revealed a significant main effect of caffeine ($F(1,35) = 19.033$, $p < 0.001$, caffeine < no caffeine), no significant main effect of taurine ($F(1,35) = 0.020$, $p = 0.887$), but no significant interaction between caffeine and taurine ($F(1,35) = 0.261$, $p = 0.613$) on the distance traveled in the light zone of the light-dark test (Fig. 7A). A two-factor ANOVA indicated a significant main effect of caffeine ($F(1,35) = 30.364$, $p < 0.001$, caffeine < no caffeine), no significant main effect of taurine ($F(1,35) = 0.122$, $p = 0.729$), but no significant interaction between caffeine and taurine ($F(1,35) = 0.639$, $p = 0.430$) on the number of entries to the light zone of the light-dark test (Fig. 7B). A two-factor ANOVA indicated a significant main effect of caffeine ($F(1,35) = 20.088$, $p < 0.001$, caffeine < no caffeine), no significant main effect of taurine ($F(1,35) = 0.024$, $p = 0.877$), but no significant interaction between caffeine and taurine ($F(1,35) = 0.137$, $p = 0.714$) on the number of crossings in the tank from one compartment to the other in the light-dark test (Fig. 7C). A two-factor ANOVA revealed no significant main effect of caffeine ($F(1,35) = 1.188$, $p = 0.283$), no significant main effect of taurine ($F(1,35) = 2.107$, $p = 0.155$), and no significant interaction between caffeine and taurine ($F(1,35) = 2.961$, $p = 0.094$) on the time spent in the light zone of the light-dark test (Fig. 7D). A two-factor ANOVA revealed no significant main effect of caffeine ($F(1,35) = 2.426$, $p = 0.128$), no significant main effect of taurine ($F(1,35) = 1.411$, $p = 0.243$), and no significant

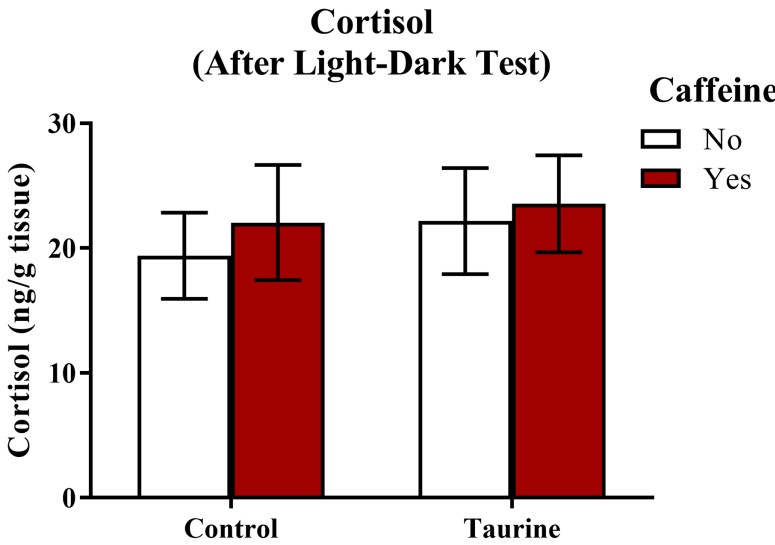

**Figure 8 Measures of zebrafish neuroendocrine function after the light-dark test.** Acute exposure to energy drink components (caffeine, taurine, or both) did not alter whole-body cortisol levels of zebrafish of fish in Experiment 2 (fish were sacrificed after the light-dark test). Bars indicate means of each group ±SEM.                                

interaction between caffeine and taurine ($F(1,35) = 1.320$, $p = 0.258$) on the latency of the fish to re-enter the light zone after the first time visiting the dark zone in the light-dark test (Fig. 7E).

Similar to the results found in the novel tank test in Experiment 1, caffeine-treated fish generally appeared to be less exploratory, as caffeine-treated fish traveled less in the light zone, made fewer crossings, and entered the light zone fewer times. Taurine generally did not alter overall exploration, and it does not appear that taurine has any mitigating or additive effects on caffeine-induced alterations in exploratory behavior in the light-dark test when both drugs were administered at the same time.

### Whole-body cortisol levels post-light-dark test

A neurochemical marker of anxiety was determined by analyzing whole-body cortisol levels of each subject (Fig. 8). A two-factor ANOVA revealed no significant main effect of caffeine ($F(1,35) = 0.243$, $p = 0.625$), no significant main effect of taurine ($F(1,35) = 0.274$, $p = 0.604$), and no significant interaction between caffeine and taurine ($F(1,35) = 0.024$, $p = 0.879$) on whole body cortisol levels (Fig. 8). Thus, it does not appear that acute exposure to different components of energy drinks altered stress hormone responses, at least when cortisol was sampled immediately after behavioral measures were assessed in the light-dark test.

## DISCUSSION

The purpose of this study was to identify, in a zebrafish model, the various behavioral and neurochemical changes elicited in response to three treatments compared to control: caffeine, taurine, and caffeine and taurine in combination. Based on previous studies, it was expected that caffeine treatment would elicit anxiogenic effects, taurine treatment

would elicit anxiolytic effects, and exposure to both caffeine and taurine would result in mixed effects on cortisol levels and behavioral measures associated with anxiety.

The results of this study indicate there are mixed effects of drug treatment on the various behavioral measures; however, in general, it appears that caffeine is anxiogenic in both the novel tank and light-dark tests. Taurine does not appear to have anxiolytic effects on its own, nor does it significantly impact the effects of caffeine when the two drugs are administered simultaneously, at least at the dose and time tested in the current study. In addition, there is no effect of drug treatment on whole-body cortisol levels in zebrafish.

The neurochemical analysis performed after each of the behavioral tests suggests that acute exposure to caffeine, taurine, or both does not affect basal levels of whole-body cortisol. Although previous studies have indicated that caffeine can alter basal levels of cortisol, the modulating effects of taurine on cortisol responses are only evident in response to acute stress (*Mezzomo et al., 2019*). Thus, the differences between the current and previously published studies are likely to be an outcome of methodological differences between experiments, such as varying durations of treatment and timing of behavioral measurements. Although behavioral alterations persist in response to acute caffeine exposure (*Tran et al., 2017*), the time course of the cortisol response may not necessarily parallel behavioral alterations induced by pharmacological agents or stressors. In adult zebrafish, whole-body cortisol levels peak at 15 min in response to acute stressors (*Ramsay et al., 2009*; *Tran, Chatterjee & Gerlai, 2014*). In the current study, cortisol was assessed 15 min after introduction to the behavioral task; thus, it is possible that any perturbations in the cortisol levels elicited by the drug treatments may have returned back to basal levels by the time of the assessment. One previous study assessed cortisol levels immediately after recording behavioral measurements and demonstrated that a dose and exposure time to caffeine comparable to the one used in the current study (100 mg for 15 min) significantly elevated whole-body cortisol levels compared to zebrafish not exposed to caffeine (*Rosa et al., 2018*). Further studies should investigate whether caffeine, taurine, and a mixture of these two drugs alter basal levels of cortisol immediately after drug exposure or if these compounds alter peak cortisol responses after exposure to an acute stressor, such as a 2-min net chase or exposure to conspecific alarm pheromone. The current results indicate that acute exposure to the different treatments do not appear to elicit longer-term alterations (i.e., 30 min after the beginning of drug exposure) in basal levels of cortisol. Alternatively, these treatments may not significantly affect basal cortisol at all, as has been observed with salivary cortisol levels in human participants (*Giles et al., 2012*).

With regards to the behavioral assays, the only treatment to significantly impact behavior was caffeine alone. Caffeine treatment significantly decreased the distance traveled, increased resting time, and increased scototaxis (light avoidance) in the light-dark test and decreased exploration of the top zone in the novel tank test. These findings support previous studies that indicate caffeine induces a heightened anxiety-like state (*Egan et al., 2009*; *Richendrfer et al., 2012*; *Rosa et al., 2018*; *Schnörr et al., 2012*; *Steenbergen, Richardson & Champagne, 2011*). Taurine treatment alone did not appear to

influence anxiety-like behavior, as measures on the different behavioral parameters did not reach statistical significance. These findings are similar to the results from a previously published study that demonstrated that taurine treatment alone had no measurable effect on anxiety-like behavior in the novel tank test (*Mezzomo et al., 2016*). In that same study, however, 1 h of exposure to taurine did alter scototaxis in the light-dark task, as subjects spent more time in the lit portion of the tank, suggesting an anxiolytic effect in this behavioral test with longer exposure to treatment (*Mezzomo et al., 2016*). In the current study, a shorter exposure time of 15 min was used to keep the taurine treatment time equivalent to the caffeine treatment. Thus, taurine alone may directly modulate anxiolytic behavior, but only in certain behavioral paradigms with a minimum exposure time of greater than 15 min. To the best of our knowledge, a full time course of the anxiolytic effects of taurine in either behavioral task has yet to be investigated. Perhaps an even longer exposure time (>1 h) would be sufficient to elicit behavioral effects in the novel tank test as well. A potential confound to this study may be the feeding regimen of the zebrafish. Currently, there is no standardized diet or feeding regimen across zebrafish colonies (*Watts et al., 2016*). However, a recent study suggests that feeding zebrafish once per day is associated with decreased exploratory behavior compared to fish that were fed twice per day (*Dametto et al., 2018*). Although all of the fish in the current study were fed similarly, it is possible that any potential effects of treatment may have been masked by anxiety induced by the feeding regimen.

The current study is the first to examine the potential interaction of taurine and caffeine on anxiety-like measures. At least at the dose and time course of taurine exposure used in the current study, it does not appear that taurine mitigates the anxiogenic effects of caffeine when subjects are exposed to both drugs simultaneously. Further studies should investigate whether a longer exposure time to taurine would mitigate the effects of caffeine on anxiety-like behavior, and, if so, what neural mechanism would best explain these effects. As taurine exposure has been shown to be anxiolytic in other studies in the literature, potentially, caffeine and taurine could modulate anxiety-like behavior via similar neural targets. Some shared molecular targets include adenosine and γ-amino butyric acid (GABA). Adenosine receptors are involved with modulating anxiety in humans, rodents, and zebrafish (*López-Cruz et al., 2017*; *Maximino et al., 2011*; *Prediger, Batista & Takahashi, 2004*; *Prediger et al., 2006*; *Vincenzi, Borea & Varani, 2017*; *Yamada, Kobayashi & Kanda, 2014*). Caffeine antagonizes adenosine receptors (*Ribeiro & Sebastião, 2010*). Specifically, blockade of adenosine $A_1$ receptors in zebrafish attenuates the anxiogenic effect of caffeine (*Maximino et al., 2011*). Higher doses or longer exposure to taurine could potentially elicit an increase in brain levels of adenosine (*Rosemberg et al., 2010*), which may partially overcome the caffeine antagonism of adenosine receptors. Alternatively, modulation of GABA transmission may be a possible mechanism by which caffeine and taurine could regulate anxiety-like behavior. The inhibitory activity of GABA is directly associated with anxiety-like responses (for review, see *Nuss, 2015*). Both caffeine and taurine may be moderating the activity of GABAergic cells, as administration of caffeine blocks GABAergic inhibitory postsynaptic potentials (*Isokawa, 2016*) and taurine enhances the activity of GABAergic interneurons (*Sava et al., 2014*). Although there is

evidence to suggest that both caffeine and taurine can mediate opposing functions within the same neurotransmitter system (e.g., adenosine or GABA), and thus have the potential to modulate anxiety by a shared circuit, it is entirely possible that each of these compounds could be modulating separate systems to alter anxiety-like behavior. Future studies should investigate whether caffeine and taurine are working on either (or both) of these putative regulators of anxiety-like behavior, or if one or both of these compounds modulate some other system entirely, such taurine modulating the glycine system (*Zhang & Kim, 2007*). Further studies should also investigate the impact of this acute regulation of neural targets on downstream effects of the HPA/HPI system.

Future studies should also address the impact of caffeine, taurine, and mixed drug exposure on different strains of zebrafish; some strains, such as the leopard strain, appear to have higher baseline levels of anxiety (*Egan et al., 2009*). Future studies should also investigate whether these compounds affect behavioral and neuroendocrine responses differently in male and female fish. Stress-related behavior may vary in zebrafish depending on sex, as has been observed in many other species (*Donner & Lowry, 2013*). Sex differences in exploratory and other behavioral responses in zebrafish have been studied less than other species (*Ampatzis & Dermon, 2016*), but indicate that sex may be a major factor in responsiveness to pharmacological manipulations (*Singer et al., 2016*).

Although the current study did not demonstrate that 15 min of exposure to taurine modulated caffeine-induced anxiety-like behavior, it is the first to study this question. More studies are required to fully elucidate any synergistic effect of caffeine and taurine, as has been observed in other measures, and whether activity of the HPA/HPI axis is involved with the regulation of anxiety-like behavior. In addition, many more studies are needed to determine whether anxiety-like states are altered by human consumption of highly caffeinated beverages with significant taurine concentrations (e.g., energy drinks). It is also important to note that energy drinks often contain many more additives that may or may not alter the properties of the two compounds under investigation of the current study. Also, given the previous literature, whether taurine supplementation would be a viable avenue for treating anxiety conditions in humans should be a focus of future investigations.

## CONCLUSIONS

The current study is the first to investigate a possible interaction between caffeine and taurine on anxiety-like behavior and neuroendocrine measures in zebrafish. Although caffeine elicited anxiogenic effects in two different behavioral paradigms, taurine treatment alone or in combination with caffeine did not significantly affect anxiety-like behavior. None of the treatments in the current study altered whole-body cortisol levels in zebrafish. However, other studies in the literature suggest that taurine may have the potential to modulate anxiety-like states with longer exposure times. Further studies are necessary to investigate the involvement of the hypothalamic-pituitary-adrenal/ interrenal axis and neurotransmitter systems such as adenosine and GABA in regulating

anxiety-like behavior altered by caffeine and taurine. In addition, supplemental products commonly consumed by humans should be investigated in more detail, particularly for those individuals at higher risk for stress or anxiety-related disorders.

## ACKNOWLEDGEMENTS

The authors would like to thank Amy Aponte, Beatriz Castro, Emma DiPasquo, Tye Dominguez, Alyssa Fassoth, Brianna Henning, Aleesa Parker, Summer Pattison, Adeel Shafiq, Jessica Singh, Horace Townsend, Elijah Weathersby, and Jennifer Wright for their technical assistance with some aspects of these studies. The use of Ethovision XT software (Noldus) was made possible by the Faculty for Undergraduate Neuroscience (FUN) Equipment Loan program.

### Funding

This work was supported by the IU Northwest Faculty Grant-in-aid of Research and the Faculty for Undergraduate Neuroscience (FUN) Equipment Loan program. The funders had no role in study design, data collection and analysis, decision to publish, or preparation of the manuscript.

### Grant Disclosures

The following grant information was disclosed by the authors:
IU Northwest Faculty Grant-in-aid of Research.
Faculty for Undergraduate Neuroscience (FUN) Equipment Loan program.

### Competing Interests

The authors declares that they have no competing interests.

### Author Contributions

- Alia O. Alia conceived and designed the experiments, performed the experiments, analyzed the data, authored or reviewed drafts of the paper, approved the final draft.
- Maureen L. Petrunich-Rutherford conceived and designed the experiments, performed the experiments, analyzed the data, contributed reagents/materials/analysis tools, prepared figures and/or tables, authored or reviewed drafts of the paper, approved the final draft.

### Animal Ethics

The following information was supplied relating to ethical approvals (i.e., approving body and any reference numbers):

Zebrafish (*Danio rerio*) are considered to be a species exempt from the Animal Welfare Act (AWA) and, thus, do not need oversight by the Institutional Animal Care and Use Committee (IACUC) at the author's university if the project is not funded by Public Health Services (PHS).

## Data Availability

The raw data are available in a Supplemental File.

## Supplemental Information

Supplemental information for this article can be found online at http://dx.doi.org/10.7717/peerj.7546#supplemental-information.

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
