# Peer review of "Anxiety-like behavior and whole-body cortisol responses to components of energy drinks in zebrafish (Danio rerio)"

_PeerJ, doi:10.7717/peerj.7546_

## Round 0.1 · original submission · Major Revisions

The reviewers prepared several comments for your attention, and I have attached a copy of this report below. The reviewers have done an excellent job. I urge you to pay special attention to the aspects raised regarding the methodology and statistics used (and how the results were interpreted). I believe that all comments will improve your manuscript quality.

·

Basic reporting

In the MS entitled “Anxiety-like behavior and whole-body cortisol responses to components of energy drinks in zebrafish (Danio rerio)” the English language is clear, and the tables and figures are easy to understand. The topic approached in the MS is relevant because the consumption of taurine and caffeine association in energy drinks is very popular, and the synergic effects of these psychoactive are still unknown.

The MS addresses current and relevant references on the subject, however, the results found in the study do not support the main idea discussed along the MS (see next evaluation topics).

Basically, some points should be rethought:

-The introduction is extensive and approaches the isolated effects of caffeine and taurine, however, little is discussed about the effects of these drugs in association. What is known about this association? This topic should be deepened.

-Which is the prevalence of the energy drinks consumption? How these energy drinks are commonly consumed (association with alcohol)? It can be related to which social problems?

-The hypotheses described in the end of introduction section are unnecessary.

Experimental design

The MS is original and is within the scope of the PeerJ. The research question is relevant and well defined, and the methodology used is referenced. However, the methodology applied to evaluate the effects of taurine and caffeine on anxiety-like behaviors can be mistaken or with poor reproducibility. It can be a problem to replicate this protocol in the future.

In relation to experimental design, some points should be considered:

-Was this study approved by an ethical committee? If yes, which is the approval protocol number?

-Mezzomo et al. (2016) showed that TAU exposure did not alter anxiety responses using the novel tank test. Anxiety-like responses trigger by taurine were observed only when the light dark paradigm was used. Why the authors choose to use the novel tank test to evaluate the synergic effects of taurine and caffeine?

-I suggest that the authors perform an additional experiment using the light-dark test in the same context, in a manner to increase the robustness of the findings.

Validity of the findings

The findings of the MS are inconclusive and without statistical significance to support the conclusions argued along all the MS.

Some points should be minutely evaluated:

-The main conclusion of the MS is that taurine exposure act mitigates the anxiety-producing effects of caffeine. However, the MS data not support this affirmation because do not exist statistical significance in the parameters discussed. No parameter evaluated in the novel tank was statistically altered by the association of taurine and caffeine. The MS only argue about trends, so, the results are not robust. I suggest that the experimental design be rethought and the light-dark test be done to confirm the data presented so far.

-The statistical analyse applied in the MS is incorrect. Two factors influence the results: taurine and caffeine, so the statistical analyse correct to be use is the two-way ANOVA.

-Why is the number of animals used in the experiments so dissimilar? In the novel tank test the number of fish per group was 10-38, and in cortisol analyse 19-39 fish per group. Please, explain why this number is so varied.

-Rosa et al. (2018) showed that acute caffeine exposure in the same concentration used in the present study (100 mg/L) increased the cortisol levels in zebrafish. However, the levels of cortisol were not altered in this study. How is possible justify this finding?

-Mezzomo et al. (2016) exposed the fish to 400 mg/L of taurine for 1 hour before evaluate anxiety-like behaviors. Why the authors perform taurine exposure for 15 minutes? There are previous studies that justify the use of this time of taurine exposure to evaluate anxiety behavior?

-In the study, taurine decreased the distance travelled when compared to other groups? However, this result is not approached in the discussion section. How this result can be explained?

-In the discussion section, many results that are not statistically significant are discussed. Mechanisms about the results are hypothesized and some points justified based on trends. Additional experiments are needed to support this discussion.

Additional comments

The topic of the study is original and relevant in neuroscience. The study has a clear objective, however, some methodological aspects need to be improved for the conclusions to be supported.

·

Basic reporting

Dear Editor

This manuscript investigated anxiety-like behavior and whole-body cortisol responses to components of energy drinks in zebrafish (Danio rerio). Overall, the manuscript is well written, the methodological approach is sound, and the authors have done a nice job.
Some methodological aspects and study design should be better described to improve clarity. The major issues are listed below:

Experimental design

1) In most studies using zebrafish the animals are fed 2-3 times a day. In this study, the animals were fed only once a day. Why this food restriction? Could this diet be involved in malnourished animals?

2) Is a week of habituation enough for the animals to get used to the new environment?

3) What is the zebrafish lineage used in the study? Different types of lineage present different behavioral profiles. The wild-type is most used being representative of population.

4) The sex effect on behavioral outcomes was not considered. There are several articles in the literature showing that males and females zebrafish may respond differently to certain situations. This could have been considered by the authors.

5) What is the density of animals per liter used in the experiment? A higher density of 5 animals per liter can alter the behavior of the animals making them more anxious.

6) Why were the animals tested on only one behavioral apparatus? Why was the light/dark teste not done?

7) I suggest that measures of zebrafish freezing behavior and measures of zebrafish motor activity be presented in graph form.

8) What are the limitations of the study?

Validity of the findings

1) A major point is about randomization and blinding. The author did not describe that randomization was performed to allocate the "treatment" groups. What was the method used for random allocation in the treatment groups? Were experimenters blind to treatment? Were data analysts blind to treatment? These questions need to be addressed and clearly stated in the methods section.

2) How was the sample size calculated? Was it based on a pilot experiment to determine effect sizes? How were the calculations performed? being that the study presents different numbers of subjects between the groups. The n is too varied which makes it possible to type I and type II errors. Why were the experimental groups unbalanced?

3) If animals considered outliers in behavior were removed because the same animals were not removed from the cortisol analysis, knowing that these animals had behavioral problems. What was the criterion for determining the outliers?

4) Did the authors verify the potential effect of the tank? Ideally, at least two tanks should be available for each treatment.

5) Control animals spent less time in the upper zone of the tank in this study compared to other articles using the same apparatus. Would be these animals more anxious even without any intervention?

6) On the results expressed as percentage that did not explore the top of the aquarium, what measure used to realize that percentage? Number of entries or time in the upper zone? And what is the significance and advantage of this analysis in relation to the other measures that were not statistically significant? This measure was confused and not usually used in the literature. The criteria commonly used to verify a alteration in exploratory behavior would be the number of crossings between the zones and the time in the upper and bottom zones of the tank. I suggest that present the data of number of crossings and time in the bottom zone of the tank, since there was no change in the time in the upper zone. If these are significant, a alteration in exploratory behavior and type-anxious effect can be observed. If the choice is to maintain this analysis, it should be better explained and substantiated.

7) Was reported that caffeine decreased the exploratory behavior of the animal in the upper zone of the tank and that this decrease in exploratory behavior is indicative of the anxiety-like effect of caffeine. However, representative data for alteration in exploratory behavior and indication of caffeine anxiety-like (time in the bottom zone, time and duration of immobility and crossings) were not statistically significant. Can say that there was actually an effect of caffeine on these behavioral parameters? I suggest that the discussion of the results should be better addressed following the results found.

8) In Figure 1 is reported values are mean ± SEM of 10-38 fish per group. The maximum number of excluded animals reported were 2 per group, the volumes expressed in the graphs were between 10-38 animals? Why were the experimental groups unbalanced?

9) Average velocity, measured in units of distance per unit time (typically, meters per second), is the average distance traveled during some time interval. If the time is the same, the distance is proportional to average speed. Therefore, table 1 should present the same ratio between the mean speed and distance data. The authors have to revise all data because it is not correct.

---

## Round 0.2 · Minor Revisions

Dear Dr. Petrunich-Rutherford,

The manuscript has considerably improved. However the reviewers raised some questions as detailed in their reviews.

I would also like to point out a specific issue: average velocity, measured in units of distance per unit time (typically, meters per second), is the average distance traveled during some time interval. If the time is the same, the distance is proportional to average speed. Therefore, the figure 1 must present the same graph independent of the values of the “y” axis. The author has to revise all data because probably it is not correct.

Kind regards,
Angelo

·

Basic reporting

The authors accepted the reviewers' suggestions and this improved the quality of the manuscript.

Experimental design

The authors explained the inquiries required by the reviewers and did an additional experiment to increase the robustness of the data, which improved the quality of the manuscript and the reliability of the data.

Validity of the findings

The authors explained all the questions required and altered the MS according to this. Mainly, they changed the statistical analysis to the correct form and thus the results of the manuscript became more robust. However, my only doubt is: Why did the authors not use a post-test of multiple comparisons after the application of two-way anova? The post-test could demonstrate some statistical significance not seen by the two-way anova interactions. I suggest that a post-test of multiple comparisons is still applied to the data.

Additional comments

The authors improved the quality of the manuscript by performing additional experiments, altering the statistical analysis and improving the writing of the manuscript.

·

Basic reporting

Dear Editor

After the first review of the article, the authors answered the questionings abouth some methodological aspects and study design and described with more clarity the results and discussion. Overall, the revised manuscript is significantly improved and well written. The methodological approach is suitable and sufficiently addressed each of the reviewers’ concerns.

However, some doubts that remained are listed below:

Experimental design

Mean speed, measured in units of distance per unit time (typically, meters per second), is the mean distance traveled during some time interval. If the time is the same, the distance is proportional to average speed. However, in figure 1, the data is conflicting and do not present the same ratio between mean speed and distance data.

In the Light-Dark Test some measurements can be difficult to obtain when the animal is in the black compartment. Can measures of total distance traveled (cm), mean speed (cm/s), immobility duration (s) be analyzed with Behavior Cloud motion-tracking software? I suggest quantifying the number of crossings and latency for first entry in the light zone.

In the cortisol assay the samples from Experiment 1 were allowed to dry at room temperature under a fume hood until the ether layer was fully evaporated; samples from Experiment 2 were dried with a light stream of air. Why this difference?

Validity of the findings

About de answer in Reviewer #1, comment #9. " In Experiment 1 (Novel Tank Test), several groups of undergraduate students collected the data". Were more experiments performed for the control group? Were the data grouped? Data of different experiments can't be grouped. Please, explain it in details.

---

## Round 0.3 · accepted · Accept

I congratulate the authors for the nice work. In this present form, the manuscript can be accept in PeerJ.